# Engineered Shellac Beads-on-the-String Fibers Using Triaxial Electrospinning for Improved Colon-Targeted Drug Delivery

**DOI:** 10.3390/polym15102237

**Published:** 2023-05-09

**Authors:** Yaoyao Yang, Wei Chen, Menglong Wang, Jiachen Shen, Zheng Tang, Yongming Qin, Deng-Guang Yu

**Affiliations:** School of Materials & Chemistry, University of Shanghai for Science and Technology, 516 Jungong Road, Shanghai 200093, China; 202342985@st.usst.edu.cn (W.C.); 191370148@st.usst.edu.cn (M.W.); 2035052318@st.usst.edu.cn (J.S.); 2035052723@st.usst.edu.cn (Z.T.); 2035052320@st.usst.edu.cn (Y.Q.)

**Keywords:** colon-targeted drug delivery, zero-order release, modified triaxial electrospinning, beads-on-the-string, core–sheath structure

## Abstract

Colon-targeted drug delivery is gradually attracting attention because it can effectively treat colon diseases. Furthermore, electrospun fibers have great potential application value in the field of drug delivery because of their unique external shape and internal structure. In this study, a core layer of hydrophilic polyethylene oxide (PEO) and the anti-colon-cancer drug curcumin (CUR), a middle layer of ethanol, and a sheath layer of the natural pH-sensitive biomaterial shellac were used in a modified triaxial electrospinning process to prepare beads-on-the-string (BOTS) microfibers. A series of characterizations were carried out on the obtained fibers to verify the process–shape/structure–application relationship. The results of scanning electron microscopy and transmission electron microscopy indicated a BOTS shape and core–sheath structure. X-ray diffraction results indicated that the drug in the fibers was in an amorphous form. Infrared spectroscopy revealed the good compatibility of the components in the fibers. In vitro drug release revealed that the BOTS microfibers provide colon-targeted drug delivery and zero-order drug release. Compared to linear cylindrical microfibers, the obtained BOTS microfibers can prevent the leakage of drugs in simulated gastric fluid, and they provide zero-order release in simulated intestinal fluid because the beads in BOTS microfibers can act as drug reservoirs.

## 1. Introduction

The development of advanced nanomaterials depends not only on the composition of the matrix but also on the external shape and internal structure [1,2]. By changing the external shape and internal structure of nanomaterials, there is potential to change the properties of the nanomaterials and develop novel functional materials [3]. The beads-on-the-string (BOTS) fibrous morphology is composed of the two most common shapes (fiber and particle) of functional nanomaterials, i.e., particles are arranged axially along the fibers [4,5]. BOTS fibers can effectively integrate the advantages of fibers and particles and realize the unification of external shape and internal structure [6,7].

Electrospinning has attracted much attention because it can prepare fibers in one step [8,9,10,11,12]. The process has the virtue of easy operation, excellent economy, and good physical properties of nanoproducts [13,14]. In the electrospinning process, the composition of the fibers can be adjusted by changing the working fluids [15,16,17]. The external shape and internal structure of the fibers can be adjusted by changing the spinneret structure [18,19]. Therefore, electrospinning can stably prepare fibers with composite shapes and structures [20], such as core–sheath [21,22,23,24,25,26], Janus [27,28,29], tri-layer core–sheath, and tri-layer Janus [30,31,32,33]. The types of electrospun fibers are increasing. BOTS fibers are one of the many derivatives of electrospun nanofiber [34,35,36]. Currently, BOTS fibers have been used in drug delivery [37,38,39,40], air filtration [41,42], superhydrophobic materials [43,44], and oil–water separation [45,46]. However, most of the studies on BOTS fibers have stayed at the stage of preparing monolithic fibers by single-fluid electrospinning [47,48,49]. There are few studies on the preparation of BOTS fibers with composite structures by multi-fluid electrospinning [50].

With the continuous development of the nano era, it has gradually become a major direction of study to apply nanomaterials in drug delivery systems [51]. Electrospun fibers have been broadly applied in drug delivery [52,53,54,55,56,57]. This is because the drug in the fibers exists in an amorphous form, which effectively improves the solubility and bioavailability of insoluble drugs. Currently, drug delivery systems have developed into many types, which include colon-targeted release, zero-order release, biphasic release, and delayed release. Oral colon-targeted drug delivery offers patients better comfort and higher compliance than injections and suppositories for the treatment of colonic disease. Oral colon-targeted delivery facilitates the efficient delivery of drugs sensitive to acid and enzymes. Among them, electrospun nanoproducts for colon-targeted release are typically pH-sensitive delivery systems, which are achieved in two main parts [58,59,60]. First, the pH of the gastrointestinal tract changes [61] and, second, the use of pH-sensitive materials. The pH-sensitive materials used in oral colon-targeted drug delivery are insoluble in acidic conditions, but they can be dissoluble in neutral or alkaline conditions [62,63]. By adding pH-sensitive materials to the matrix or using pH-sensitive materials as a sheath layer, the inactivation of acid-sensitive drugs can be avoided and premature drug release can be reduced [64]. Polymeric materials of natural origin are popular as drug carriers for biomedical applications [65,66]. Shellac is a natural biomaterial. It can generate strong intermolecular hydrogen bonds by carboxyl groups under acidic conditions, thus preventing its dissolution. As the pH of the gastrointestinal tract increases, the shellac will swell and dissolve. Therefore, shellac is often used as a matrix or as coated tablets in colon-targeted drug delivery [67,68].

Currently, most studies have used monolithic fibers for colon-targeted drug delivery, which often show burst release during the early release stage because the drug is inevitably present on the surface of those fibers [69,70]. Therefore, the preparation of core–sheath electrospun fibers using pH-sensitive materials encapsulated in a drug-loaded matrix is an ideal method to solve this problem [71,72]. Furthermore, by changing the composition, external shape, internal structure, and relative size of the fibers, desirable and controllable colon-targeted drug release profiles can be obtained.

This study unifies matrix composition, external shape, and internal structure and proposes a new method for the stable preparation of microfibers with a BOTS shape and core–sheath structure by multi-fluid electrospinning. The BOTS nanofiber with a composite structure was prepared by triaxial electrospinning using hydrophilic polyethylene oxide (PEO) [73] and the water-insoluble anti-colon cancer drug curcumin (CUR) [74] in the core layer, ethanol in the middle layer, and the natural pH-sensitive biomaterial shellac in the sheath layer. Subsequently, the core–sheath BOTS microfibers were evaluated with a series of characterizations to analyze the differences between the colon-targeted drug delivery properties of BOTS and linear cylindrical microfibers.

## 2. Materials and Methods

### 2.1. Materials

Shellac (CAS No.: 9000-59-3) and PEO (Mw = 300,000) were obtained from Shanghai Macklin Biochemical Technology Co., Ltd. (Shanghai, China). CUR (purity ≥ 95.0%), ethanol, phosphate-buffered saline (PBS), hydrochloric acid (HCl), and Tween 80 were sourced from Sinopharm Chemical Reagent Co., Ltd. (Shanghai, China). Chemicals were of analytical grade and were used directly.

### 2.2. Electrospinning

The core fluid (PEO–CUR) was obtained from 0.5 g PEO and 0.05 g CUR dissolved in 10 mL of 80% aqueous ethanol. The middle fluid was ethanol. The sheath fluid was obtained from 3 g shellac dissolved in 10 mL of ethanol. In the above three fluids, the middle and sheath fluids are unspinnable.

The prepared fluids were loaded into syringes, respectively. An 18 G syringe needle and homemade coaxial and triaxial spinnerets were used for electrospinning. The homemade electrospinning systems were composed of a high-voltage power supply (ZGF 60 kV/2 mA, Hua-Tian, Wuhan, China), a collector (aluminum-foil-wrapped cardboard box), and syringe pumps (KDS100, Cole-Parmer, IL, USA). The electrospinning/electrospraying parameters are shown in Table 1. The ambient environmental temperature and the relative humidity were 20 ± 4 °C and 40 ± 8%.

### 2.3. Shape and Structure Characterization

The samples to be tested were sputter-coated with gold for 150 s in a vacuum atmosphere to make them conductive. The shape of the fibers was characterized by scanning electron microscopy (SEM, Quanta450FEG, FEI, OR, USA). No fewer than 100 random points were selected in the SEM images for measuring the size distribution and average diameter of the fibers using ImageJ software (National Institutes of Health, MD, USA). The structure of the fibers was characterized using transmission electron microscopy (TEM, H7600, Hitachi, Tokyo, Japan) after the electrospun fibers were loaded on the copper mesh.

### 2.4. Physical Form and Compatibility Characterization

The samples were investigated with an X-ray diffractometer (XRD, Bruker-AXS, Karlsruhe, Germany) using a Cu Kα ray as a light source at 40 kV and 30 mA. The samples were analyzed in the range of 2θ from 10° to 60°. The samples were tested after KBr compression treatment using a Fourier transform infrared spectrometer (FTIR, Spectrum 100, PerkinElmer, MA, USA) in the spectral scanning range of 4000–500 cm^−1^ and a resolution of 2 cm^−1^.

### 2.5. In Vitro Drug Release

First, 50 mg of the drug-loaded nanofibrous membrane was dissolved in 225 mL of 0.05% (*v*/*v*) Tween 80 in HCl solution (pH = 2, simulated gastric fluid) for 2 h. Then, the membrane was transferred to 225 mL of 0.05% (*v*/*v*) Tween 80 in PBS solution (pH = 7.4, simulated intestinal fluid) for 8 h. Dissolution experiments were carried out in an oscillating incubator apparatus (SHZ-88, Shuibei Science Experimental Instrument Factory, Changzhou, China) at 37 °C ± 1 °C and 50 rpm. At predetermined time intervals, 4 mL of the dissolution medium was withdrawn and 4 mL of fresh medium was added to maintain a constant volume of the dissolution medium. The sample absorbance was tested at λ_max_ = 425 nm, and the amount of Cur released was calculated through the calibration curve. The experiments were repeated six times, and the results are reported as mean ± SD.

Three kinetic models served as a method to estimate the drug-release mechanism from the nanofibrous membrane, where *k*_0_, *k*_1_, and *k*_p_ are the constants in zero-order kinetics (Equation (1)), first-order kinetics (Equation (2)), and Peppas models (Equation (3)), respectively. *Q* and *Q*_0_ are the cumulative and initial drug release at time *t*. *n* is the diffusion coefficient in the Peppas model.
(1)$$Q=k_{0}t+Q_{0}$$
(2)$$Q=Q_{0}{(1-e}^{{-k}_{1}t})$$
(3)$$Q=k_{p}t^{n}$$

### 2.6. Drug Loading and Encapsulation Efficiency

To free all the loaded CUR, 50 mg of fibrous membrane was dissolved into 5 mL of ethanol. The above sample was diluted to 225 mL in simulated intestinal fluid. Then, 4 mL of the supernatant was removed, and the absorbance of the sample was tested. Based on Equations (4) and (5), the drug loading (*DL*) and encapsulation efficiency (*EE*) was calculated.
(4)DL%=WAWm ×100
where *W_A_* is the actual mass of CUR in the nanofibrous membrane and *W_m_* is the mass of the nanofibrous membrane.
(5)EE%=WAWT ×100
where *W_T_* is the theoretical mass of the CUR in the nanofibrous membrane.

## 3. Results and Discussion

### 3.1. Triaxial Electrospinning and Its Implementation Process

Electrospinning takes advantage of the easy interaction between the electrostatic energy and the working fluid [75,76,77]. When coaxial electrospinning is used to prepare electrospun fibers, there is additional contact friction and viscous resistance between the fluid interface of the core and sheath layers. Therefore, an interfacial shear between the fluids is formed. The coaxial electrospinning can prepare core–sheath fibers when the shear stress is greater than the interfacial tension between the fluids [78,79]. The sheath fluid of conventional coaxial electrospinning must be spinnable. In contrast, it is essential that the core fluid of modified coaxial electrospinning is spinnable. Modified coaxial electrospinning has the advantage of expanding the range of materials used for electrospinning, improving the ability to handle different types of working fluids and stabilizing the electrospinning process [80]. In this study, linear cylindrical electrospun microfibers with a core–sheath structure were prepared using a sheath fluid of 30% shellac (which is not spinnable) and a core fluid of PEO–CUR (which is spinnable). On this basis, this study innovatively used ethanol as a middle fluid to change the interfacial tension between the core–sheath fluid and the prepared BOTS electrospun microfibers with a core–sheath structure by modified triaxial electrospinning (Figure 1).

The homemade triaxial spinneret and the corresponding electrospinning process are shown in Figure 2. Figure 2a,b show the photograph and schematic diagram of the homemade triaxial spinneret. It is composed of three concentric metal capillaries with internal to external diameters of 0.9, 1.8, and 2.8 mm, respectively (Figure 2c). In addition, the core metal capillary protrudes approximately 0.2 and 0.4 mm from the middle and sheath (Figure 2d). The design reduces the contact and diffusion between different fluids and promotes the formation of the core–sheath structure. The connection between the spinneret and the syringe is shown in Figure 2e. In this case, the syringe with the middle fluid is linked directly to the spinneret, and syringes with the core and sheath fluid are linked to the core and sheath sides of the spinneret via silicone tubes. Figure 2f–h show the electrospinning process for F1, F4, and F7 fibers, respectively. The F1 and F4 fibers have a typical electrospinning process. When ethanol was used as the middle fluid, the straight jet of F7 had more bifurcations compared to F1 and F4. This is probably due to ethanol changes in the interfacial tension between the core–sheath fluids, which affects the electrospinning process.

### 3.2. Shape and Structure

Figure 3 shows the SEM images of the nanoproducts prepared by single-fluid electrospinning/electrospraying. The sheath fluid (30% shellac) was used to prepare P1 nanoparticles by electrospraying, which have an overall spherical shape (Figure 3a). The particle size of P1 shows a nonuniform distribution, and its average diameter is about 1.03 ± 0.58 µm. The core fluid (PEO–CUR) was used to prepare F1 nanofibers by single-fluid electrospinning; they have a linear cylindrical shape (Figure 3b). There are no observable beads on the F1 nanofibers, and their average diameter is approximately 0.70 ± 0.09 µm.

The SEM images of the microfibers that were prepared by modified coaxial/triaxial electrospinning are shown in Figure 4. F2–4 microfibers were prepared by modified coaxial electrospinning with a gradual increase in the flow rate of the sheath (Figure 4a–c). Although a few spindles could be observed on the F3 and F4 microfibers, the overall linear cylindrical shape was maintained. Core–sheath linear cylindrical microfibers can be formed if the unspinnable sheath fluid is uniformly distributed on the core fluid surface during the bending and whipping area of the modified coaxial electrospinning. However, as the sheath fluid flow rate increases, the drying time required for the charged jet increases, which makes part of the sheath fluid gradually shrink under the surface tension to form spindles. The average diameters of F2, F3, and F4 are gradually increased when increasing the fluid flow rate of the sheath fluid, which is 1.59 ± 0.22, 1.86 ± 0.20, and 1.93 ± 0.22 µm, respectively.

The microfibers with a BOTS shape (F5, F6, F7) were prepared by modified triaxial electrospinning using a middle fluid of ethanol (Figure 4d–f). Compared to F2–F4 microfibers prepared with the same sheath fluid, F5–F7 microfibers significantly promoted BOTS shape formation due to the presence of ethanol in the middle layer. This is because the middle layer of ethanol is able to change the interfacial tension between the core and sheath fluids. The average diameters (linear fiber segment) of F5–F7 microfibers are 1.32 ± 0.19, 1.46 ± 0.22, and 1.45 ± 0.24 µm, respectively. To verify the difference between BOTS fibers and linear cylindrical fibers, F7 microfibers with more beads and the F4 microfibers were selected for subsequent experiments.

Figure 5 shows the TEM images of F7. Figure 5a shows the bead segment of F7. Figure 5b,c show a partially enlarged view of Figure 5a. From the TEM images of Figure 5b,c, it can be observed that the junction of the fiber and the bead has a core–sheath boundary, and the sheath thickness is about 0.21 ± 0.01 µm. It was deduced that the beads also had a core–sheath structure. The fiber segment of F7 is shown in Figure 5d. It can be clearly observed from the TEM image that the fiber has a core–sheath structure, and the sheath thickness is about 0.22 ± 0.01 µm.

Electrospinning/electrospraying enables one-step and stable preparation of nanoproducts. In the above two processes, the BOTS is usually treated as a byproduct. Therefore, studies usually focus on the elimination of the BOTS by optimization experiments. However, the novel external shape and internal structure of BOTS fibers can be used to develop new functional nanomaterials.

During the preparation of nanoproducts by single-fluid electrospinning, nanoparticles, BOTS fibers, and linear cylindrical fibers can be obtained, respectively, as the viscosity of the working fluid increases (Figure 6a). The modified coaxial electrospinning is able to prepare monolithic BOTS fibers by using solvent as the sheath fluid [81]. When the sheath fluid is a small molecule polymer, BOTS core–sheath electrospun fibers can be prepared [82]. This is because the small molecule fluid in the sheath is unable to uniformly encapsulate the core fluid during the electrospinning process. Subsequently, the sheath fluid gradually shrinks under the interfacial tension, and it forms beads with a core–sheath structure. However, the fiber segment of the fibers is a monolithic structure (Figure 6b). This study innovatively used solvent as the middle fluid in the triaxial electrospinning. It effectively isolates the core and sheath fluids and changes the interfacial tension between the core and sheath fluids. Moreover, as the flow rate of the unspinnable sheath fluid increases, the number of beads gradually increases. This method makes it possible to stably prepare BOTS fibers with a core–sheath structure (Figure 6c).

### 3.3. Physical Form

Figure 7 shows the physical form of the raw materials (CUR, PEO, and shellac) and fibers (F1, F4, and F7) assessed by XRD patterns. The shellac is an amorphous material because it lacks sharp Bragg reflections in the XRD pattern. PEO is a semi-crystalline material, which has two distinctive peaks at about 2θ = 19° and 2θ = 23°. The CUR has many distinct sharp peaks, indicating that it is a crystalline material. F1, F4, and F7 retained the characteristic peaks of PEO at about 2θ = 19° and 2θ = 23°, and no other peaks were observed. The results indicated that most of the crystal structure of PEO was transformed into amorphous forms after the electrospinning process. No characteristic peaks of CUR were observed in F1, F4, and F7 fibers, indicating that CUR exists in the drug-loaded fibers in an amorphous form.

### 3.4. Compatibility between the Drug and Matrix

In Figure 8, the compatibility of the fibers was assessed by the FTIR spectra. The FTIR spectra of CUR have peaks at 3510, 1603, 1510, and 1154 cm^−1^, corresponding to hydroxyl (-OH) stretching, stretching vibrations of the benzene ring, C=C vibrations, and ether group (C-O-C) stretching, respectively. The carbonyl (C=O) characteristic peak of CUR corresponds to 1628 cm^−1^ between 1620 cm^−1^ and 1650 cm^−1^ [83]. PEO has a peak at 2877 cm^−1^, indicating the presence of C-H, and the triple characteristic peaks at 1145, 1092, and 1059 cm^−1^ indicate the presence of C-O-C. The shellac has peaks at 3432, 2931, and 1716 cm^−1^, corresponding to -OH, -CH, and C=O, respectively [84].

The typical characteristic peaks of CUR in the FTIR spectra of F1 shifted and even disappeared. This phenomenon indicates that CUR and PEO form a molecular composite, i.e., there is good compatibility between them. The characteristic peaks of CUR in F4 and F7 also shifted and disappeared. The main characteristic peaks of the shellac were retained, which implied the formation of the core–sheath structure. The molecular formula of the raw materials is shown in Figure 8. It indicates that hydrogen bonds may be formed between C=O in CUR as a proton acceptor and -OH in PEO as a proton donor. Hydrophobic interactions may occur between the benzene ring in the CUR and the long carbon chain of the PEO. These secondary interactions can effectively promote compatibility between the drug and matrix, thus improving their homogeneity and stability.

### 3.5. In Vitro Colon-Targeted Drug Release

The DL of F1, F4, and F7 fibers is 8.84 ± 0.06%, 1.66 ± 0.02%, and 1.66 ± 0.02%, respectively. The EE of F1, F4, and F7 fibers is 97.30 ± 0.63%, 98.12 ± 0.98%, and 97.95 ± 1.21%, respectively. The encapsulation efficiency results showed that there was almost no drug loss in the preparation of fibers by electrospinning. This is because electrospinning prepares solid dispersions by the “bottom-up” method, which has the advantage of rapid drying of drugs. It can effectively encapsulate the drug into the matrix of electrospun fibers.

Figure 9a shows the relationship between the CUR relative release (%) and time (h). The fibers were first placed in simulated gastric fluid for 2 h, and then in simulated intestinal fluid for 8 h. The cumulative CUR release of F1, F4, and F7 was 96.58 ± 2.19%, 97.61 ± 1.45%, and 98.07 ± 1.84%, respectively. Among them, F1 nanofibers released 93.12 ± 2.65% of CUR in the first 1 h with a significant burst release curve. F4 and F7 microfibers have typical colon-targeted drug release profiles. They had relatively similar CUR releases in the first 2 h of 7.38 ± 0.51% and 8.38 ± 0.22%, respectively. The insolubility of shellac as a sheath layer effectively avoids drug leakage under simulated gastric fluid. The F4 and F7 microfibers did not show burst release in the early stage of the simulated intestinal fluid. The dissolution of shellac after water absorption and swelling is relatively slow, resulting in sustained release profiles for F4 and F7 in simulated intestinal fluid.

The time (h) required for the certain CUR relative release (%) is shown in Figure 9b. The time required to release 20%, 40%, 60%, 80%, and 90% of CUR is 3.36 h, 5.33 h, 6.57 h, 8.19 h, and 9.20 h, respectively. When the CUR relative release is the same, F4 requires less time. This is because the beads of F7 microfibers can fully encapsulate the core layer of PEO-CUR, which takes a long time to dissolve the shellac. There are more beads that act as drug reservoirs; thus, the F7 microfibers provide better-sustained release under simulated intestinal fluid.

To visualize the role of the beads in the F7 microfibers, a simple dissolution test was performed. The F4 and F7 nanofibrous membranes were removed after 1 h and 3 h in simulated intestinal fluid and dried to observe the shape (Figure 9c–j). The F4 was in simulated intestinal fluid for 1 h, and the fibers changed from linear cylindrical to curved elliptical (Figure 9c). The F4 deformed after absorbing water, swelling, and drying. Irregular pores could be noticed on the surface of the F4 due to the gradual dissolution of the shellac (Figure 9d). The F4 deformed more obviously after 3 h release in simulated intestinal fluid (Figure 9e). More importantly, some of the microfibers showed folds and collapses (Figure 9f). This is due to the dissolution of the core layer, and after drying the microfibers gradually collapse and eventually form folds. After 1 h and 3 h, F7 microfibers also showed pores and collapses (Figure 9g–j). It has been noticed that the beads in the F7 microfibers were able to maintain their original shape without collapse. The conjecture that the shellac in the beads could adequately encapsulate the core layer and the bead could act as a drug reservoir was verified. Therefore, it can provide a zero-order drug release curve in simulated intestinal fluid.

The drug release profiles of F4 and F7 microfibers under simulated intestinal fluid were fitted using three kinetic models, and the results are shown in Table 2. It can be seen from the correlation coefficient R^2^ that F4 and F7 follow the zero-order kinetic model. In particular, the correlation coefficient $R^{2}=0.9900$ of F7 microfibers perfectly follows the zero-order kinetic model. The zero-order release curve is used as the most desirable sustained release. It can eliminate the burst release and provide sustained release, keeping a stable drug concentration in the blood within the therapeutic window. The mechanism of drug release from the microfibers was assessed by the Peppas model. The equations for F4 and F7 microfibers are $Q_{4}=5.61t^{1.27}$($R^{2}=0.9560$) and $Q_{7}=3.85t^{1.42}$ ($R^{2}=0.9828$). Their diffusion coefficients were much larger than 0.89 (1.27 and 1.42), indicating that the CUR was released from the drug-loaded microfibers mainly by the skeleton corrosion mechanism.

### 3.6. Drug Release Mechanism of the Fibers

Most traditional drug delivery systems control the release mechanism by selecting different matrices. However, with the continuous development of nanomaterials, the preparation of complex nanostructures enables attempts to provide advanced drug delivery systems. The drug release mechanisms of F1, F4, and F7 fibers are shown in Figure 10. The burst release of water-insoluble CUR from the F1 nanofibers in the simulated gastric fluid is due to the water-soluble matrix PEO. The sheath layer of shellac is a pH-sensitive biomaterial that leads a small number of CUR molecules to be released from the F4 and F7 microfibers in the simulated gastric fluid. With the dissolution of shellac, CUR molecules were gradually released from core–sheath linear F4 microfibers in simulated intestinal fluid. The core–sheath BOTS F7 microfibers also prevented the release of CUR molecules in simulated gastric fluid. The beads in F7 act as a drug reservoir in simulated intestinal fluid to prevent premature leakage of CUR, thus providing a better-sustained release. Therefore, F7 microfibers provide a zero-level release profile in simulated intestinal fluid. Compared with core–sheath linear fibers, core–sheath BOTS fibers can be considered a novel material. Thus, it provides an interesting idea for the development and application of nanomaterials. Based on the protocols reported here, the capability of multi-fluid electrospinning in tailoring the shape and inner structure of fibers [85,86,87,88] can be further combined with new therapeutical strategies and new active/inert components in the literature [89,90,91,92,93,94,95] for developing more medicated materials.

## 4. Conclusions

In the modified triaxial electrospinning, the middle fluid (ethanol) was used to isolate the core fluid (5% PEO–0.5% CUR) and the sheath fluid (30% shellac). This changed the interfacial tension between the core and sheath fluid and promoted the formation of BOTS microfibers. SEM and TEM images indicate that the microfibers had a BOTS shape and core–sheath structure. XRD and FTIR results showed that the CUR had an amorphous form, and it had good compatibility with the PEO. The in vitro drug release curve demonstrated that BOTS microfibers of F7 had a preferable colon-targeted drug release curve compared to the linear cylindrical microfibers of F4. The F7 had little drug leakage in the simulated gastric fluid and a zero-order release curve in the simulated intestinal fluid. Therefore, BOTS fibers can effectively improve the oral colon-targeted delivery of drugs. The BOTS fibers effectively integrate fibers and particles, and they complete the organic unification of external shape and internal structure. This paves a brand-new way to develop novel functional materials.

## Figures and Tables

**Figure 1 polymers-15-02237-f001:**
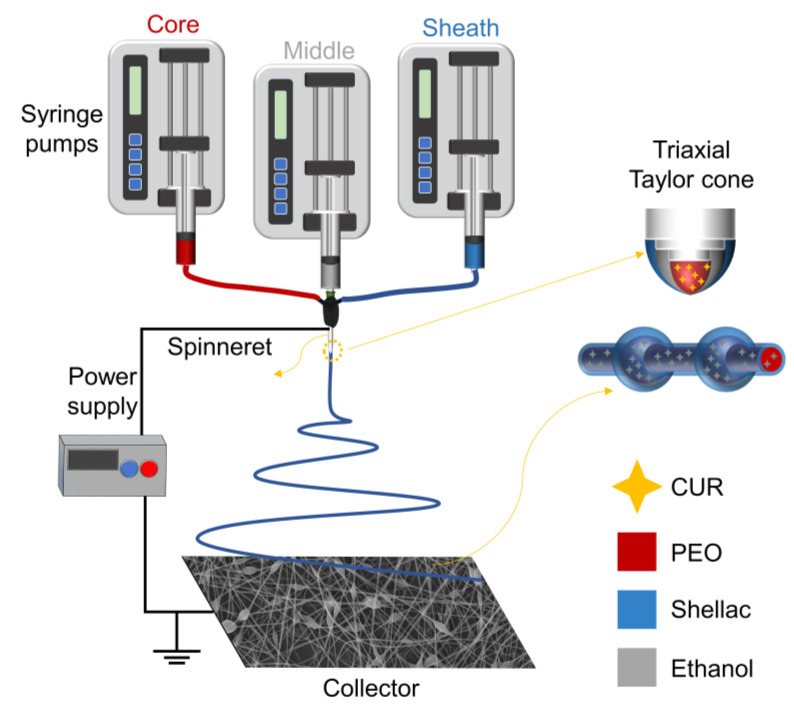
Schematic diagram of the modified triaxial electrospinning process.

**Figure 2 polymers-15-02237-f002:**
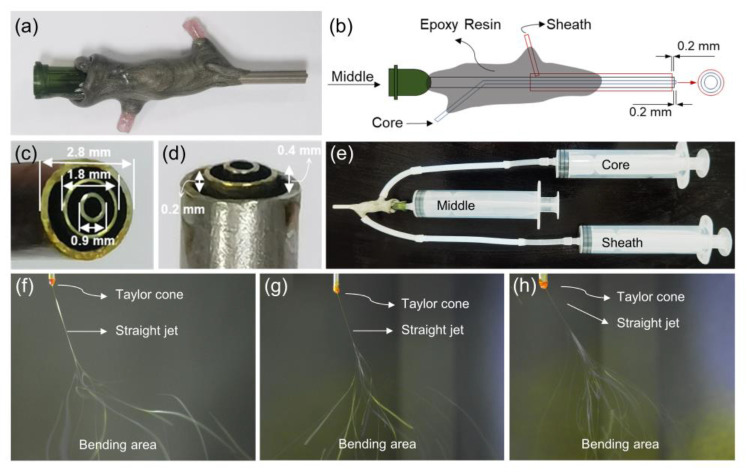
Homemade triaxial spinneret and electrospinning implementation process. (**a**,**b**) Photograph and schematic diagrams of homemade triaxial spinneret; (**c**,**d**) photographs of the spinneret tip; (**e**) the connection between spinneret and syringe; (**f**–**h**) the electrospinning process for F1, F4, and F7 fibers.

**Figure 3 polymers-15-02237-f003:**
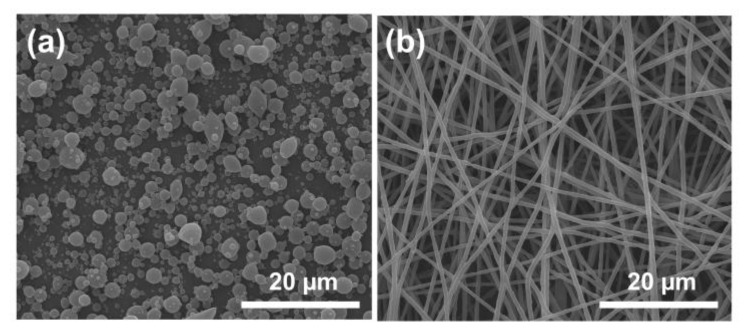
The SEM images of nanoproducts prepared by electrospinning/electrospraying. (**a**) P1; (**b**) F1.

**Figure 4 polymers-15-02237-f004:**
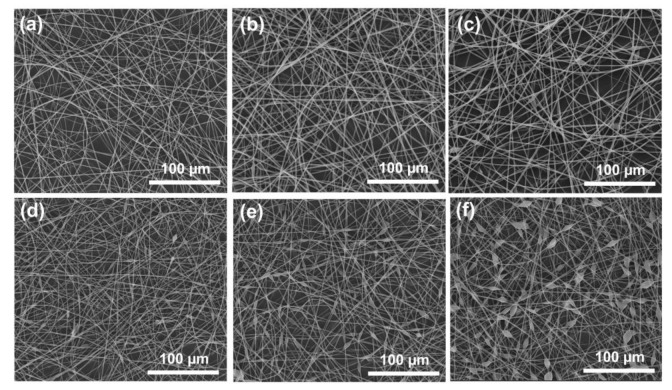
The SEM images of the microfibers that were prepared by modified coaxial/triaxial electrospinning. (**a**) F2; (**b**) F3; (**c**) F4; (**d**) F5; (**e**) F6; (**f**) F7.

**Figure 5 polymers-15-02237-f005:**
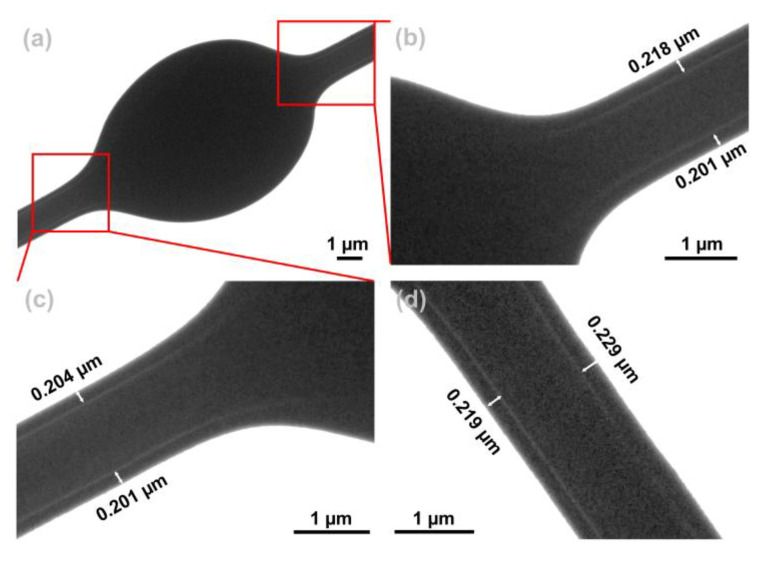
TEM images of F7. (**a**) The fiber and bead junction of F7; (**b**,**c**) partially enlarged view of (**a**); (**d**) the linear fiber segment of F7.

**Figure 6 polymers-15-02237-f006:**
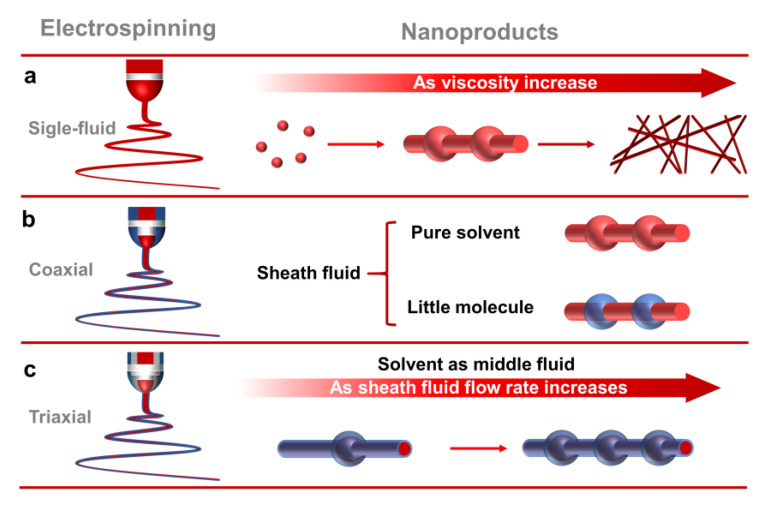
Mechanism of BOTS fibers formation. (**a**) Single-fluid electrospinning; (**b**) coaxial electrospinning; (**c**) triaxial electrospinning.

**Figure 7 polymers-15-02237-f007:**
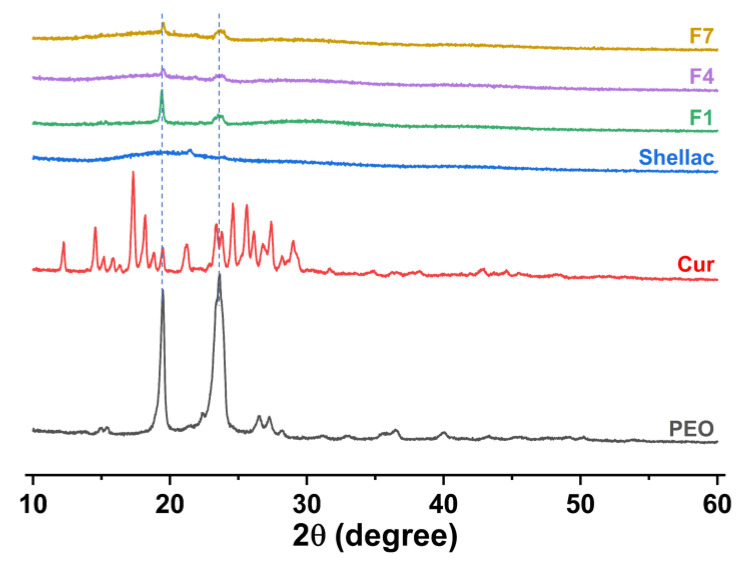
The raw materials and fibers assessed by XRD patterns.

**Figure 8 polymers-15-02237-f008:**
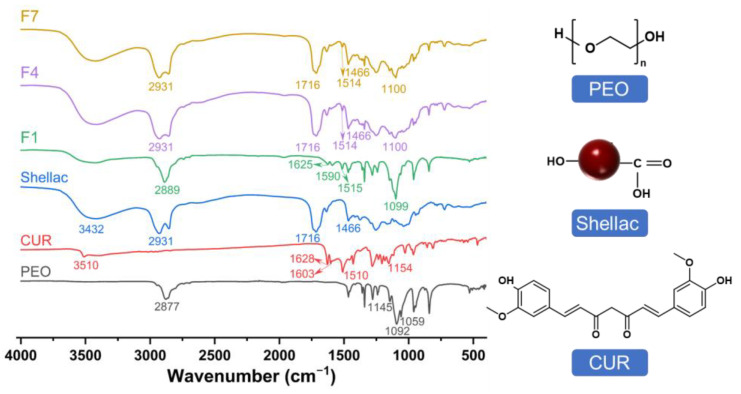
The compatibility of the fibers was assessed by the FTIR spectra and molecular formula of the raw materials.

**Figure 9 polymers-15-02237-f009:**
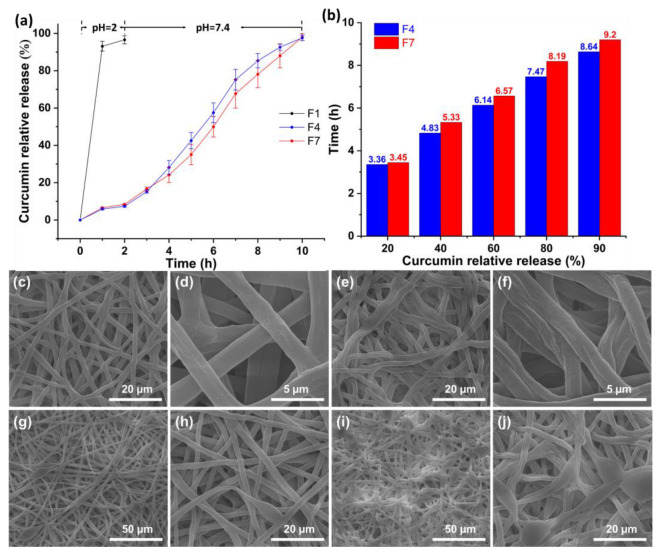
(**a**) Colon-targeted drug release profiles of fibers; (**b**) the time required for the certain CUR relative release of F4 and F7 microfibers; SEM images of F4 and F7 microfibers after drug release under simulated intestinal fluid: (**c**,**d**) F4 after 1 h; (**e**,**f**) F4 after 3 h; (**g**,**h**) F7 after 1 h; (**i**,**j**) F7 after 3 h.

**Figure 10 polymers-15-02237-f010:**
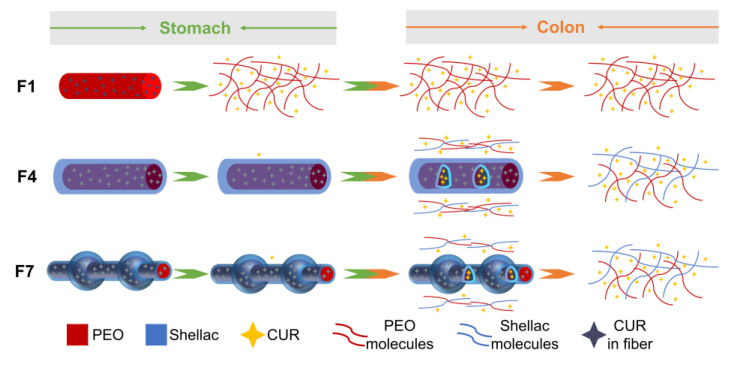
The mechanisms of drug release from F1, F4, and F7 fibers.

**Table 1 polymers-15-02237-t001:** Parameters of the electrospinning/electrospraying.

Process	No.	Fluid Flow Rate (mL/h)			Applied Voltage (kV)	Collecting Distance (cm)	Shape/ Structure
		Core ^1^	Middle ^2^	Sheath ^3^			
Electrospraying	P1	-	-	0.6	9.5	15	Particles/ Monolithic
Single	F1	0.6	-	-	6		Linear/ Monolithic
Coaxial	F2	0.6	-	0.2			Linear/ Core–sheath
	F3	0.6	-	0.4			
	F4	0.6	-	0.6			
Triaxial	F5	0.6	0.2	0.2		20	BOTS ^4^/ Core–sheath
	F6	0.6	0.2	0.4			
	F7	0.6	0.2	0.6			

^1^ The core fluid is 5% PEO–0.5% CUR. ^2^ The middle fluid is ethanol. ^3^ The sheath fluid is 30% shellac. ^4^ BOTS represent beads-on-the-string.

**Table 2 polymers-15-02237-t002:** Fitting drug release profiles for F4 and F7 microfibers under simulated intestinal fluid using three kinetic models.

No.	Zero-Order		First-Order		Peppas Model		
	$k_{0}$	$R^{2}$	$k_{1}$	$R^{2}$	$k_{p}$	$R^{2}$	n
F4	12.44	0.9728	0.02	0.9163	5.61	0.9560	1.27
F7	12.34	0.9900	−0.10	0.9716	3.85	0.9828	1.42

## Data Availability

All the data generated by this study are contained within the manuscript, so they are not deposited in a public repository.

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
