# Peer review of "Engineered Shellac Beads-on-the-String Fibers Using Triaxial Electrospinning for Improved Colon-Targeted Drug Delivery"

_polymers, 2023, doi:10.3390/polym15102237_

Round 1
Reviewer 1 Report
The overall impression of the manuscript is very good. However, I have a few comments.
First of all, I recommend minor improvements and editing of the English language. It still can be "polished" a bit more. Some of the mentioned issues can be found in my comments.
Row 97 “The middle fluid is the solvent of ethanol” – ??? – Please explain what does it mean? Maybe "solution"?
In the characterization part for some of the used equipment, the model and the manufacturer were mentioned but for some were not. Please unify the text.
I must admit, I like the description of the homemade triaxial spinneret. I`m pretty sure it will be used by readers. Also the sample marking is simple and useful: P1 – particles, F1 – fibers…
However, fibers with a diameter above 1 µm are not nanofibers anymore, they may be considered microfibers, but not nano. Please change it in the whole manuscript.
For fibers F4-F7 what part of the fiber was measured to obtain the average diameter? The bead or the thin fibrous part? Please mention both numbers with respective notes.
Row 373 “solvent of ethanol has been used” – maybe the solution of ethanol?
I recommend accepting this manuscript after fixing the mentioned minor issues.
Comments are listed above.
Author Response
Response to Reviewer 1 Comments
The overall impression of the manuscript is very good. However, I have a few comments.
Point 1: First of all, I recommend minor improvements and editing of the English language. It still can be "polished" a bit more. Some of the mentioned issues can be found in my comments.
Response 1: The manuscript has been checked and polished by native English-speaking colleagues.
Point 2: Row 97 “The middle fluid is the solvent of ethanol” – ??? – Please explain what does it mean? Maybe "solution"?
Response 2: Thank you for your reminder. The middle fluid is ethanol, it is a solvent. We changed “The middle fluid is the solvent of ethanol” to “The middle fluid is ethanol” to prevent misunderstanding.
Point 3: In the characterization part for some of the used equipment, the model and the manufacturer were mentioned but for some were not. Please unify the text.
Response 3: We have unified the text in the model and the manufacturer of the used equipment.
Point 4: I must admit, I like the description of the homemade triaxial spinneret. I`m pretty sure it will be used by readers. Also the sample marking is simple and useful: P1 – particles, F1 – fibers… However, fibers with a diameter above 1 µm are not nanofibers anymore, they may be considered microfibers, but not nano. Please change it in the whole manuscript.
Response 4: Thank you for catching the mistakes. We have changed the nanofibers to microfibers in the full manuscript.
Point 5: For fibers F4-F7 what part of the fiber was measured to obtain the average diameter? The bead or the thin fibrous part? Please mention both numbers with respective notes.
Response 5: We measured the diameter of the fibrous part of microfibers and annotate it in the manuscript.
Point 6: Row 373 “solvent of ethanol has been used” – maybe the solution of ethanol?
Response 6: Thank you very much for your patience and reminder again. We changed “solvent of ethanol has been used” to “the middle fluid (ethanol) has been used” to prevent misunderstanding.
Point 7: I recommend accepting this manuscript after fixing the mentioned minor issues.
Response 7: Thank you for reviewing this manuscript carefully and for your positive comments.
Reviewer 2 Report
The authors present a study titled “Engineered Shellac Beads Using Tri-Axial Electrospinning for An Improved Colon-Targeted Drug Delivery”. The findings showed that ethanol employed as an intermediary fluid in modified triaxial electrospinning facilitated the growth of BOTS nanofibers and changed the interfacial tension between the core and sheath fluids. Compared to the linear cylindrical nanofibers of F4, the BOTS nanofibers of F7 had a better colon-targeted drug release curve. F7 has a zero-order release curve into simulated intestinal juice and little drug leakage into simulated gastric juice. Although the provided results appear to be conclusive some comments need to be addressed as mentioned below.
· Further experiments are required to support this study such as disease activity index, MTT viability assay using colorectal cells, biocompatibility, PCR and western blot analysis etc. Some of them can be conducted by the authors.
· What was the payload of the 50 mg nanofibrous membrane used for drug release?
· To support the stability of NFs, SEM images of prepared nanofibers within different time periods along with distribution histogram can be provided.
· The author should discuss the route of drug administration to make a clear understanding of this investigation.
· The SEM images in Figure 9 can be measured on two particular scale bars. (e,f), (g,h) and (i,j) can be measured as in (c,d).
No specific comments for English
Author Response
Response to Reviewer 2 Comments
The authors present a study titled “Engineered Shellac Beads Using Tri-Axial Electrospinning for An Improved Colon-Targeted Drug Delivery”. The findings showed that ethanol employed as an intermediary fluid in modified triaxial electrospinning facilitated the growth of BOTS nanofibers and changed the interfacial tension between the core and sheath fluids. Compared to the linear cylindrical nanofibers of F4, the BOTS nanofibers of F7 had a better colon-targeted drug release curve. F7 has a zero-order release curve into simulated intestinal juice and little drug leakage into simulated gastric juice. Although the provided results appear to be conclusive some comments need to be addressed as mentioned below.
Point 1: Further experiments are required to support this study such as disease activity index, MTT viability assay using colorectal cells, biocompatibility, PCR and western blot analysis etc. Some of them can be conducted by the authors.
Response 1: We agree with your comments that further experiments are necessary. Your suggestion provides a direction for our next research. Unfortunately, due to the limited time, we did not supplement further experiments. In this study, we aim to the preparation of core-sheath beads-on-the-string fibers by triaxial electrospinning and explore the drug release mechanism of beads-on-the-string fibers. So we only conducted a preliminary study on the process-shape/structure-application relationship of core-sheath beads-on-the-string fibers.
Point 2: What was the payload of the 50 mg nanofibrous membrane used for drug release?
Response 2: The theoretical drug loading of 50 mg F1 fibrous membrane was 9.10% (4.55 mg). The actual loading and encapsulation efficiency of the F1 fibrous membrane for drug release was 8.84 ± 0.06% and 97.30 ± 0.63%. The theoretical loading of 50 mg F4 and F7 fibrous membranes was 1.70% (0.85 mg). The actual loading and encapsulation efficiency of the F4 fibrous membrane for drug release was 1.66 ± 0.02% and 98.12 ± 0.98%. The actual loading and encapsulation efficiency of the F7 fibrous membrane for drug release was 1.66 ± 0.02% and 97.95 ± 1.21%.
Point 3: To support the stability of NFs, SEM images of prepared nanofibers within different time periods along with distribution histogram can be provided.
Response 3: SEM images of F4 and F7 fibers after 1 h and 3 h in the simulated intestinal fluid are shown in Figure 9c-j. The stability of the fibers is preserved during the 1 h and 3 h. However, as the shellac of the sheath layer dissolves, the fibers become adherent after drying making it impossible to visualize the changes in fiber morphology. Therefore, the SEM images of the subsequent time periods are not provided in Figure 9.
Point 4: The author should discuss the route of drug administration to make a clear understanding of this investigation.
Response 4: We added the advantages of the oral route of colon-targeted drug administration in the introduction (Lines 58-61). We add the advantages of beads-on-the-string fibers for oral colon-targeted drug administration in the conclusion (Line 384).
Point 5: The SEM images in Figure 9 can be measured on two particular scale bars. (e,f), (g,h) and (i,j) can be measured as in (c,d).
Response 5: We used the same scale bars for (e, f) and (g, h) because the smaller scale bars can observe the changes in the morphology of F4 fibers in detail. The same scale bars were used for (i, j) and (c, d) because the larger scale bars can easy observation of larger diameter beads.
Reviewer 3 Report
Journal: Polymers
Manuscript ID: polymers-2366653
Title: Engineered Shellac Beads Using Tri-Axial Electrospinning for An Improved Colon-Targeted Drug Delivery
In this manuscript, the authors report the development of electrospun polymeric fibers with bead-on-string morphology as efficient drug vehicles for the controlled and colon-targeted delivery of curcumin. Using a homemade triaxial spinneret, they prepared core-sheath fibers composed of polyethylene oxide/curcumin in the core and shellac in the sheath layer, comprising an ethanol solution as the middle fluid in the triaxial spinneret to facilitate the bead-on-string fiber architecture. The fabricated fibers allowed for the controlled delivery of curcumin in simulated gastric fluid, indicating the efficiency of beads in the core-sheath fibers.
Recommendation: Major revision.
Although it is an interesting work, in general the manuscript does not meet the criteria of the Journal and needs a major revision in order to be suitable for publication.
The language and grammar in the manuscript are not of a sufficient standard and need further work to improve it, as sentence structure, word choice, and spelling mistakes are found in the text. (Especially in Abstract, Introduction, and Materials and Methods sectors)
For example
Line 13: as a core fluid…as a sheath fluid… “as” sounds awkward. PEO, curcumin or shellac are not fluids..they have been used to prepare the corresponding fluids.
Line 14: the solvent of ethanol – just “ethanol” sounds better.
Line 19: and its components have good compatibility can be indicated – the sentence is awkward.
Line 23: gastric juice - gastric fluid.
Line 32: The beads-on-the-string (BOTS) is composed - The beads-on-the-string (BOTS) fibrous morphology is composed… or The beads-on-the-string (BOTS) fibrous architecture is composed.
Line 32: develop new science and technology – develop novel functional materials.
Line 34: fiber and particles arranged axially along the fiber..- Okay for particles, but how a fiber is arranged axially along the fiber??
Line 37: “it can prepare nanofibers in one step by the "bottom-up method”.. – by the bottom-up method? The sentence is awkward and needs to be rephrased.
Line 40: during the electrospinning, the composition of the nanofibers can be controlled by changing the type and number of working fluids.. Not exactly “during” the electrospinning.. the sentence needs to be rephrased.
Line 45: “They are continuously enriching the diversity of electrospun nanofibers” - the sentence needs to be rephrased.
Line 50: There are only a fewer studies have been conducted - the sentence needs to be rephrased.
Line 55: when the drug is dissolved in the matrix for electrospinning, the drug is usually distributed in the fibers in an amorphous form. - the meaning of the sentence is not clear.
Line 58: electrospinning has developed multiple advanced drug delivery - awkward sentence.
Line 63: The pH-sensitive materials are insoluble in acidic conditions, but they can be dissoluble in neutral or alkaline conditions. - Why can’t it be the opposite?
Line 65: the inactivation of acid-sensitive drugs and reduces premature drug release can be avoided – awkward sentence.
Line 66: Polymeric molecules with natural sources – polymeric materials of natural origin or from natural sources.
Line 67: Shellac is a typical natural biomaterial – what is a “typical” natural biomaterial?
Line 77: the desirable and controllable colon-targeted drug release curves – drug release profiles sound better.
Line 97: The middle fluid is the solvent of ethanol - The middle fluid was.
Line 101: are used for electrospinning – were used.
Line 104: First, the fluid flow rate was set on the syringe pumps. Then the high-voltage power supply and spinneret were connected through the alligator clip. Why these informations are important?
Line 105: Finally set the voltage and collect the nanofibers on the collector. – the sentence needs to be rephrased.
Line 133: At predetermined intervals, remove 4 mL of the sample from the dissolution medium and add the same volume of fresh medium to ensure a constant volume of the dissolution. The sentence needs to be rephrased.
Line 144: Dissolve 50 mg nanofibrous membrane into 5 mL of ethanol to completely release the drug. - the sentence needs to be rephrased.
Line 194: which has - which have.
Line 197: which has - which have.
Line 198: and its average diameter - and their average diameter.
Line 213: the middle of ethanol.
Line 217: As shown in Figure 5, it is the TEM image of F7 - in Figure 5, it is shown the TEM image of F7.
…And so on………
Title: Engineered Shellac Beads – Perhaps the title should change, as it is not beads, but bead-on-string electrospun fibers.
Equations 4 and 5 are correct? Shouldn’t be DL (%) = (Wa/Wm) x 100 and EE (%) = (WA/WT) x 100 respectively..?
What is the molecular weight of shellac?
In table 1, it is not clear what is the applied voltage and collecting distance for all formulations.
What type of dissolution apparatus did the authors use?
Line 136: was calculated with the standard equation – what equation? From calibration curve?
Line 136: The measurements were performed six times – Is it the experiments that were run six times?
Line 208: average diameters of F1, F2, and F3 – most probably is F2, F3 and F4.
Figure 9: It is better to show SEM images with the same scale bar.
Why the authors have chosen to study the F1, F4 and F7 formulations? What about the F2, F3, F5 and F6..?
Since the average diameter of the fabricated fibers is in the μm scale maybe it is better to use the term microfibers or micro-/nanofibers.
The authors should cite a previous work using the triaxial spinneret for curcumin release:
Liu, Y.; Chen, X.; Gao, Y.; Liu, Y.; Yu, D.; Liu, P. Electrospun Core–Sheath Nanofibers with Variable Shell Thickness for Modifying Curcumin Release to Achieve a Better Antibacterial Performance. Biomolecules 2022, 12, 1057. https://doi.org/10.3390/biom12081057
The language and grammar in the manuscript are not of a sufficient standard and need further work to improve it, as sentence structure, word choice, and spelling mistakes are found in the text. (Especially in Abstract, Introduction, and Materials and Methods sectors)
Round 2
Reviewer 3 Report
Journal: Polymers
Manuscript ID: polymers-2366653
Title: Engineered Shellac Beads Using Tri-Axial Electrospinning for An Improved Colon-Targeted Drug Delivery
In this revised version, the authors have adequately addressed the referees’ comments and the manuscript has been improved. However, the revised manuscript is not prepared in a Track-Changes format and is difficult to understand which words or sentences have been changed or replaced. (Although the new words/sentences are highlighted in red there is no indication for the deleted words or sentences).
The authors should delete the words/sentences that have corrected/replaced and revise the manuscript with track-changes.
Recommendation: Minor revision.
Line 16: used in a modified..
Line 25: revealed
Line 47: electrospinning process
Line 52: They are increasing the types of electrospun fibers – Who are they? Perhaps it is better to use something like “over the years various types of electrospun fibers have been reported.”
Figure 9: line 359-the drug release diagrams are hiding the Sem images.
English language can be further improved